# In-Hospital Mortality and Morbidity in Cancer Patients with COVID-19: A Nationwide Analysis from the United States

**DOI:** 10.3390/cancers15010222

**Published:** 2022-12-30

**Authors:** Ziad Abuhelwa, Anas Alsughayer, Ahmad Y. Abuhelwa, Azizullah Beran, Wasef Sayeh, Waleed Khokher, Omar Sajdeya, Sadik Khuder, Ragheb Assaly

**Affiliations:** 1Department of Medicine, University of Toledo, Toledo, OH 43606, USA; 2Department of Pharmacy Practice and Pharmacotherapeutics, College of Pharmacy, University of Sharjah, Sharjah P.O. Box 27272, United Arab Emirates; 3Sharjah Institute for Medical Research, University of Sharjah, Sharjah P.O. Box 27272, United Arab Emirates; 4Division of Gastroenterology and Hepatology, Indiana University, Indianapolis, IN 46202, USA; 5School of Population Health, University of Toledo, Toledo, OH 43614, USA; 6Department of Mathematics & Statistics, College of Natural Sciences and Mathematics, University of Toledo, Toledo, OH 43614, USA; 7Division of Pulmonology and Critical Care Medicine, Department of Medicine, University of Toledo, Toledo, OH 43606, USA

**Keywords:** cancer, mortality, morbidity, COVID-19

## Abstract

**Simple Summary:**

Patients with cancer are considered a vulnerable population and might have an increased risk for severe outcomes of coronavirus disease 2019 (COVID-19). We used the Healthcare Cost and Utilization Project Nationwide Inpatient Sample (NIS) 2020 database, the largest inpatient database in the United States, to assess COVID-19 outcomes in cancer patients. We have demonstrated that cancer patients hospitalized for COVID-19 have increased odds of all-cause in-hospital mortality and acute respiratory failure compared with non-cancer patients. Lung cancer patients have been demonstrated to have the worst mortality outcome.

**Abstract:**

Background: Coronavirus disease 2019 (COVID-19) caused significant mortality and mortality worldwide. There is limited information describing the outcomes of COVID-19 in cancer patients. Methods: We utilized the Healthcare Cost and Utilization Project Nationwide Inpatient Sample (NIS) 2020 database to collect information on cancer patients hospitalized for COVID-19 in the United States. Using the International Classification of Diseases, 10th revision, Clinical Modification (ICD-10-CM) coding system, adult (≥18 years) patients with COVID-19 were identified. Adjusted analyses were performed to assess for mortality, morbidity, and resource utilization among cancer patients. Results: A total of 1,050,045 patients were included. Of them, 27,760 had underlying cancer. Cancer patients were older and had more comorbidities. The all-cause in-hospital mortality rate in cancer patients was 17.58% vs. 11% in non-cancer. After adjusted logistic regression, cancer patients had a 21% increase in the odds of all-cause in-hospital mortality compared with those without cancer (adjusted odds ratio (aOR) 1.21, 95%CI 1.12–1.31, *p*-value < 0.001). Additionally, an increased odds in acute respiratory failure rate was found (aOR 1.14, 95%CI 1.06–1.22, *p*-value < 0.001). However, no significant differences were found in the odds of septic shock, acute respiratory distress syndrome, and mechanical ventilation between the two groups. Additionally, no significant differences in the mean length of hospital stay and the total hospitalization charges between cancer and non-cancer patients. Conclusion: Cancer patients hospitalized for COVID-19 had increased odds of all-cause in hospital mortality and acute respiratory failure compared with non-cancer patients.

## 1. Introduction

Coronavirus disease 2019 (COVID-19) is an infectious disease caused by the severe acute respiratory syndrome coronavirus-2 (SARS-CoV-2) [1]. The clinical manifestations of COVID-19 are variable. Most patients are asymptomatic but some may develop severe respiratory disease that can lead to multi-organ failure and death [2]. Worldwide, until December 2022, around 6.5 million people died due to COVID-19, and of them, about 1 million are in the United States [3].

Studies showed that COVID-19 is associated with significant mortality and morbidity, especially in patients with underlying comorbidities [4]. Several risk factors have been associated with increased mortality rates in COVID-19. These risk factors include older age, male sex, African American race, and presence of hypertension, diabetes, chronic pulmonary diseases, or other comorbidities [4,5,6,7]. Patients with underlying cancer are considered vulnerable population, and at higher risk of having severe clinical course and poor prognosis due to COVID-19 [8,9].

There are no sufficient data to evaluate the mortality and morbidity of COVID-19 in patients with cancer. We aimed to evaluate whether the diagnosis of cancer is associated with a higher risk of poor outcomes in patients with COVID-19. In this study, we evaluated the differences in baseline clinical characteristics and all-cause in-hospital mortality between COVID-19 patients with and without cancer. Moreover, we looked into the morbidity and resource utilization outcomes during the hospital stay. To our knowledge, this is the largest study from the United States on the outcomes of hospitalized COVID-19 patients with cancer. We believe that the data in this study can be useful in assisting healthcare professionals in comprehending how cancer affects COVID-19 outcomes.

## 2. Methods

### 2.1. Data Source

In this retrospective cohort study, COVID-19 patients admitted to acute care hospitals in the United States were identified using the Healthcare Cost and Utilization Project Nationwide Inpatient Sample (NIS) 2020 database. NIS is the largest all-payer and publicly available inpatient database in the United States which was developed and is maintained by the Agency for Healthcare Research and Quality. The NIS approach is described in detail elsewhere [10]. Briefly, using data from the American Hospital Association’s yearly hospital survey, hospitals are grouped according to ownership/control, bed number, teaching status, and geographic region. Then, data on patients’ demographics, diagnoses, and resource use are gathered from a random 20% sample of all patients within each stratum and entered into the database. Each discharge is then weighted (weight = total number of discharges from all acute care hospitals in the United States/number of discharges included in the 20% sample) to make the NIS nationally representative.

The NIS is the nation’s most comprehensive source of hospital data, which enables researchers to study healthcare delivery and patient outcomes [11]. This database is a discharge-level database that contains deidentified clinical and nonclinical data items at the patient and hospital levels. As a result, multiple admissions for a single patient are considered separate discharges and are entered separately into the database. The International Classification of Diseases, 10th revision, Clinical Modification (ICD-10-CM) coding system is used to code patient-level information.

### 2.2. Study Subjects

Patients with a principal diagnosis of COVID-19 who have underlying cancer as secondary diagnosis were included in the analyses. The included cancers were lung, breast, colorectal, prostate, leukemia, lymphoma, and multiple myeloma. Patients were excluded if they were less than 18 years of age. The ICD-10-CM diagnosis codes were used to extract data (Appendix A). The Institutional Review Board of University of Toledo deemed the research project exempt from approval because it is a retrospective review of already collected deidentified data.

### 2.3. Study Variables and Outcomes

The primary outcome was all-cause in-hospital mortality. Secondary outcomes were (i) morbidity as measured by the rates of septic shock, acute respiratory failure, acute respiratory distress syndrome, and mechanical ventilation (ii) resource utilization as measured by length of hospital stay and total hospitalization charges. Multiple potential confounders were collected and accounted for during the analysis, including patient’s age, sex, race, admission type and day, hospital bed size, hospital teaching status, hospital region, insurance status, income in patient’s zip code, Charlson’s comorbidity index and comorbidities including congestive heart failure, coronary artery disease, chronic pulmonary disease, diabetes mellitus, hypertension, chronic renal failure, liver disease, obesity, smoking, and alcohol and drugs abuse.

Patient vital status at discharge was coded directly in NIS. Septic shock, acute respiratory failure, acute respiratory distress syndrome, mechanical ventilation, and comorbidities (congestive heart failure, coronary artery disease, chronic pulmonary disease, diabetes mellitus, hypertension, chronic renal failure, liver disease, obesity, smoking, alcohol abuse and drugs abuse) were identified using the appropriate ICD-10-CM codes (Appendix A). The length of hospital stay, total hospitalization charges and patients’ demographics were directly obtained from the NIS database.

### 2.4. Statistical Analysis

Analyses were performed using STATA (IC-17.0 version, STATA Corp). NIS is based on a complex sampling design that includes stratification, clustering, and weighting. This software facilitates analysis to produce nationally representative unbiased results, variance estimates and *p*-values. Weighting of patient-level observations was implemented to obtain estimates for the entire population in the United States of hospitalized patients with COVID-19. The Rao-Scott chi-square test was used for categorical variables and Student t-test was used for continuous variables. Multivariable regression analysis models were used to adjust the results for potential confounders. Logistic regression was used for binary outcomes (all-cause in-hospital mortality, septic shock, acute respiratory failure, acute respiratory distress syndrome, and mechanical ventilation) and linear regression was used for continuous outcomes (length of hospital stay, and total hospitalization charges). We also performed subgroup sensitivity analysis on the different types of cancers to assess all-cause in-hospital mortality. All *p*-values were two sided, with 0.05 as threshold for statistical significance.

## 3. Results

### 3.1. Patients and Hospital Characteristics

Figure 1 shows the flow diagram for study inclusion. A total of 1,050,045 patients were included in the analysis. Of them, 27,760 had underlying cancer. Table 1 summarizes patient’s and hospital’s characteristics. COVID-19 patients with cancer were older (mean age (95% Confidence Interval (CI)), 70.92 (70.52–71.32) vs. 64.57 (64.41–64.74) years, *p*-value < 0.001), had more comorbidities (99.93% had Charlson’s comorbidity index ≥ 3 vs. 42.87%, *p*-value < 0.001), and more likely to be treated at large teaching hospitals compared with non-cancer patients (70.95% vs. 68.9%, *p*-value 0.005). There were no significant differences between the two groups with regard to admission type and day, presence of underlying liver disease, and alcohol or drug abuse. Greater than 35% of patients in both groups were treated in hospitals in the South region.

### 3.2. In-Hospital Mortality

The overall all-cause in-hospital mortality rate for COVID-19 patients was 11.17% (Table 2). In COVID-19 patients with cancer, the all-cause in-hospital mortality rate was 17.58% vs. 11% in those without cancer. After adjusted logistic regression analysis, COVID-19 patients with cancer had about a 21% increase in the odds of all-cause in-hospital mortality compared with those without cancer (adjusted odds ratio (aOR) 1.21, 95%CI 1.12–1.31, *p*-value < 0.001) (Table 3).

### 3.3. Morbidity

#### 3.3.1. Septic Shock

The overall rate of septic shock in COVID-19 patients was 3.35%. The rate was 3.87% in COVID-19 patients with cancer vs. 3.33% in those without cancer (Table 2). Adjusted logistic regression analysis showed no statistically significant difference in the odds of septic shock between the two groups (aOR 0.95, 95%CI 0.8–1.12, *p*-value 0.524) (Table 3).

#### 3.3.2. Acute Respiratory Failure

The overall rate of acute respiratory failure in COVID-19 patients was 56.12% (Table 2). The rate among cancer patients was 56.93% vs. 56.1% in non-cancer patients (Table 2). Adjusted logistic regression analysis showed about a 14% increase in the odds of acute respiratory failure in patients with cancer compared with those without cancer (aOR 1.14, 95%CI 1.06–1.22, *p*-value <0.001) (Table 3).

#### 3.3.3. Acute Respiratory Distress Syndrome

The overall rate of acute respiratory distress syndrome in COVID-19 patients was 5.22%, and it was comparable among patients with and without cancer (4.99% vs. 5.22%, respectively) (Table 2). Adjusted logistic regression analysis demonstrated no statistically significant difference in the odds of acute respiratory distress syndrome between the two groups (aOR 0.93, 95%CI 0.8–1.07, *p*-value 0.31) (Table 3).

#### 3.3.4. Mechanical Ventilation

About 9.38% of all patients with COVID-19 required mechanical ventilation. Among patients with cancer, the rate of mechanical ventilation was 10.14% compared to 9.36% in patients without cancer (Table 2). Adjusted logistic regression analysis did not show a statistically significant difference in the odds of mechanical ventilation between patients with and without cancer (aOR 0.98, 95%CI 0.88–1.09, *p*-value 0.767) (Table 3).

### 3.4. Resource Utilization

#### 3.4.1. Length of Hospital Stay

The overall mean length of hospital stay in COVID-19 patients was 7.48 days, 8.07 days for patients with cancer vs. 7.46 days for those without cancer (Table 2). Adjusted linear regression analysis did not show a significant difference in mean length of hospital stay between patients with cancer and without cancer (adjusted mean difference (aMD) −0.1, 95%CI −0.39–0.18, *p*-value 0.479) (Table 3).

#### 3.4.2. Total Hospitalization Charges

The overall mean total hospitalization charges for COVID-19 patients were USD78,591. For patients with cancer, the mean total hospitalization charges were USD82,894 vs. USD78,474 in patients without cancer (Table 2). After adjusted linear regression analysis, there was no significant difference in the mean total hospitalization charges between the two groups (aMD USD136, 95%CI −5323–5594, *p*-value 0.961) (Table 3).

### 3.5. Sensitivity Analysis of Mortality Outcome by Cancer Type

A sensitivity analysis of all-cause in-hospital mortality by cancer type was conducted to assess whether the increased odds of mortality in COVID-19 patients is driven by a particular cancer type. Of the 27,760 hospitalized cancer patients for COVID-19, 4400 (15.58%) had lung cancer, 3340 (12.03%) had breast cancer, 1675 (6.03%) had colorectal cancer, 4080 (14.7%) had prostate cancer, 6230 (22.44%) had leukemia, 4320 (15.56%) had lymphoma, 3000 (10.8%) had multiple myeloma and 715 (2.58%) had two or more cancers where the latter were excluded from the sensitivity analysis. The aOR of the adjusted logistic regression analysis by cancer type is presented in Figure 2.

Patients with lung cancer (aOR 1.57, 95%CI 1.31–1.88, *p*-value < 0.001), colorectal cancer (aOR 1.45, 95%CI 1.07–1.96, *p*-value 0.015), leukemia (aOR 1.22, 95%CI 1.02–1.45, *p*-value 0.016), and lymphoma (aOR 1.30, 95%CI 1.05–1.61, *p*-value 0.016) were significantly associated with an increase in all-cause in-hospital mortality compared to non-cancer patients. On the other hand, breast cancer, prostate cancer and multiple myeloma were not associated with an increased odds in the rate of all-cause in-hospital mortality. Appendix A shows mortality, morbidity, and resource utilization outcomes according to cancer type.

## 4. Discussion

Utilizing the largest inpatient all payer database in the United States, we found that COVID-19 patients with cancer had a significantly higher rate of mortality compared with those without cancer. After adjusted analysis, the diagnosis of cancer in patients hospitalized for COVID-19 was associated with 21% increase in the odds of all-cause in-hospital mortality and 14% in the odds of acute respiratory failure. Our study showed no significant differences in the odds of septic shock, acute respiratory distress syndrome or mechanical ventilation between cancer and non-cancer patients hospitalized for COVID-19. To our knowledge, this is the largest study in the United States to evaluate the mortality and morbidity in cancer patients with COVID-19.

We found that cancer patients hospitalized for COVID-19 tend to be older and have more comorbidities. They had significantly higher rates of congestive heart failure, coronary artery disease, hypertension, chronic pulmonary diseases, and renal failure compared with those without cancer. There are several reports showed that older age and comorbidities are associated with adverse outcomes in patients with COVID-19 [4,5,6,12,13].

The all-cause in-hospital mortality rate in COVID-19 patients with cancer (17.58%) was significantly higher compared with non-cancer patients (11%). This was in line with previously reported study by Wang et al., who demonstrated an overall mortality rate of 14.93% (100/670) in cancer patients with COVID-19 compared with non-cancer patients (5.26%, 780/14,840) [14]. A large national study from Belgium included COVID-19 patients with solid malignancies showed a 30-day in-hospital mortality rate of 31.7% (283/892) in cancer patients which was significantly higher compared with non-cancer patients (20%, 1922/9594) [15]. The reported mortality rates in the literature amongst COVID-19 patients with cancer are highly variable. For example, Sharafeldin et al. and Kuderer et al. reported mortality rates of 14.8% (2888/19,515) and 13% (121/928) among patients with COVID-19 and cancer, respectively [16,17]. On the other hand, a pooled analysis, that included 2922 hospitalized patients with COVID-19 and cancer, showed a 30-day mortality rate of 30% (95%CI: 25–35%, I^2^ 82%), and 15% (95%CI: 9–22%, I^2^ 73%) in 624 patients who were treated in both inpatient and outpatient settings [18]. This difference is mortality among the inpatient and outpatient settings demonstrates the importance of primary care role in the prevention of hospitalization among cancer patients with COVID-19 [19]. The variability in the reported mortality rates can be attributed to the difference in the sample size between different studies. In addition, some studies reported 30-day mortality while others reported all-cause mortality, and the clinical setting; inpatient vs. outpatient as hospitalization itself is an index of disease severity. For these reasons, the reported mortality rates should be interpreted cautiously.

While our study showed an increase in odds of mortality in cancer patients hospitalized for COVID-19, a study by Rüthrich et al., utilized the LEOSS (Lean European Open Survey on SARS-CoV-2 infected patients) registry, showed comparable COVID-19 related deaths in patients with cancer to those without cancer (22.5%, 97/435 vs. 15%, 367/2636) after adjusting for age, sex, and comorbidities [20]. Similarly, Miyashita et al. reported no difference in the risk of death between COVID-19 patients with or without cancer (relative risk 1.15, 95%CI 0.84–1.57) [9]. This disparity between our study and what Rüthrich et al. and Miyashita et al. reported may be attributed to the larger sample size in our study.

The TERAVOLT (Thoracic Cancers International COVID-19 Collaboration), a global consortium of patients with thoracic cancers, reported a high mortality rate of 33% (66/200) in COVID-19 patients with thoracic malignancies that include lung cancer, this is higher compared with our study (20.91% in lung cancer patients) [21]. Our study showed that among different cancer types, lung cancer patients had the highest increase in the odds of all-cause in-hospital mortality rate. The higher overall mortality rate in lung cancer patients compared to other cancers may be related to the known underlying diseased lung, the likely reduced baseline pulmonary function as a result of prior lung resection surgeries or radiation therapy, and/or potential emphysematous changes as a result of smoking that may make the lungs more susceptible to negative outcomes of COVID-19.

Overall, COVID-19 is associated with a high morbidity burden on the affected patients. Our adjusted analysis showed a 14% increase in the odds of acute respiratory failure in cancer patients hospitalized for COVID-19 compared with non-cancer patients. Though no significant difference in the rate of mechanical ventilation in cancer patients compared with non-cancer patients (10.14% vs. 9.36%, respectively). In LEOSS study, 78 COVID-19 patients with cancer out of 435 (17.93%) required mechanical ventilation [20], and a report from the National COVID Cohort Collaborative reported a rate of mechanical ventilation of 8.2% (1606/19,515) among cancer patients with COVID-19 [16].

Worldwide, COVID-19 pandemic overwhelmed the healthcare systems. Though, in regard to resource utilization, our study showed no significant differences in the length of hospital stay and the total hospitalization charges in COVID-19 patients with cancer compared with those without cancer. A similar result on the length of hospital stay was reported in a study from Turkey that included patients with hematologic malignancies and COVID-19 [22]. The mean length of hospital stay among our COVID-19 patients with cancer (8.07 days) was lower than what is reported in the literature. Manzano et al. and Aboueshia et al. reported a mean length of hospital stay in COVID-19 cancer patients of 11.2 and 12.8 days, respectively [23,24].

Despite the strengths of our study owing to its large sample size and scope, there are several limitations. First, an administrative database was used to collect our data. Claims-based databases have been shown to be vulnerable to incorrectly entered or missing codes. However, less than 3% of the variables we used had missing data. Second, cancer patients, irrespective of their symptoms, are more likely to get COVID-19 testing. The percentage of COVID-19 patients who were asymptomatic could not be determined because we were unable to acquire information on symptoms at presentation. Third, due to the retrospective design of our study, the exposure is unamenable to randomization thus we relied on regression models to account for confounders. Therefore, there is a risk of residual confounding. We minimized that risk by controlling for a variety of patient and hospital-level characteristics. Fourth, the results of this study represent the outcomes of adult patients who received treatment in the United States; outcomes may vary in environments with fewer resources or regions of the world with limited access to healthcare. Finally, results are only applicable to the year 2020, and could vary thereafter, especially with the development of vaccines and other treatment options.

## 5. Conclusions

Our study demonstrated that cancer patients hospitalized for COVID-19 had higher rates of all-cause in-hospital mortality and acute respiratory failure compared with those without cancer. Patients with lung cancer, particularly, had the worst outcomes compared to other cancers. Additionally, we found no significant differences in the rates of septic shock, acute respiratory distress syndrome, mechanical ventilation, and resource utilization such as length of hospital stay and total hospitalization charges between cancer and non-cancer groups.

## Figures and Tables

**Figure 1 cancers-15-00222-f001:**
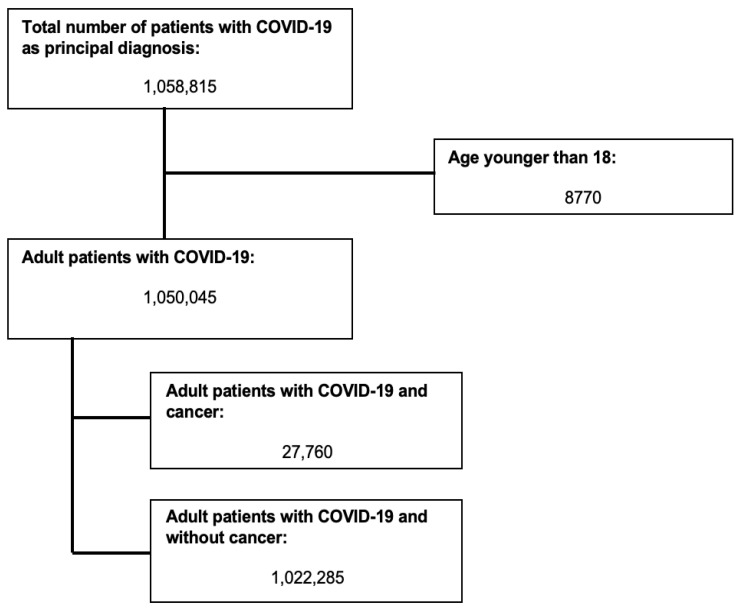
Selection of study subjects.

**Figure 2 cancers-15-00222-f002:**
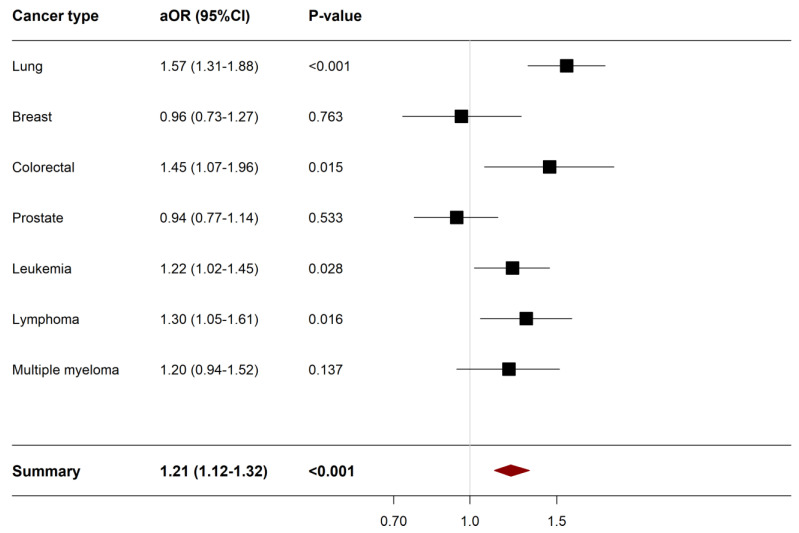
Forest plot describing the results of sensitivity analysis of all-cause in-hospital mortality according to cancer type. Abbreviations: aOR; adjusted odds ratio, CI, confidence interval, Variables were adjusted for were age, sex, race, admission type, admission day, hospital bed size, hospital teaching status, hospital region, insurance status, median annual income, and comorbidities. Statistical significance (*p*-value < 0.05).

**Table 1 cancers-15-00222-t001:** Summary of patients and hospitals characteristics.

		Cancer (*n* = 27,760)	No Cancer (*n* = 1,022,285)	*p*-Value
Age (year) mean (95%CI)		70.92 (70.52–71.32)	64.57 (64.41–64.74)	<0.001 *
Female Sex *n* (%)		12,055 (43.43%)	483,500 (47.3)	<0.001 *
Race *n* (%)	White	17,590 (63.36)	518,695 (50.74)	<0.001 *
	Black	4860 (17.51)	183,205 (17.92)	
	Hispanic	3145 (11.33)	205,905 (20.14)	
	Asian or pacific islander	535 (1.93)	32,285 (3.16)	
	Native American	115 (0.41)	10,480 (1.03)	
	Other	835 (3%)	40,800 (3.99)	
	Missing	680 (2.45)	30,915 (3.02)	
Admission type *n* (%)	Elective	880 (3.17)	30,250 (2.96)	0.384
Admission day *n* (%)	Weekend	6970 (25.11)	267,565 (26.17)	0.848
Hospital bed size *n* (%)	Small	6545 (23.58)	263,300 (25.76)	<0.001 *
	Medium	7640 (27.52)	295,960 (28.95)	
	Large	13,575 (48.9)	463,025 (45.29)	
Hospital teaching status *n* (%)	Teaching	19,695 (70.95)	704,334 (68.9)	0.005 *
Hospital region *n* (%)	Northeast	5630 (20.28)	179,970 (17.6)	<0.001 *
	Midwest	7885 (28.4)	236,640 (23.15)	
	South	10,335 (37.22)	429,056 (41.97)	
	West	3910 (14.08)	176,620 (17.28)	
Insurance status *n* (%)	Medicare	18,855 (67.92)	529,720 (51.82)	<0.001 *
	Medicaid	2035 (7.33)	119,690 (11.7)	
	Private Insurance	5625 (20.26)	284,845 (27.86)	
	Self-pay	370 (1.33)	35,430 (3.47)	
	No Charge	60 (0.22)	2.540 (0.25)	
	Other	780 (2.81)	48,070 (4.7)	
	Missing	35 (0.13)	1990 (0.19)	
Median annual income in the patient’s zip code *n* (%)	USD1–USD45,999	8305 (29.92)	345,615 (33.8)	<0.001 *
	USD46,000–USD58,999	7940 (28.6)	278,010 (27.19)	
	USD59,000–USD78,999	5985 (21.56)	220,440 (21.56)	
	USD79,000 or more	5125 (18.46)	162,090 (18.86)	
	Missing	405 (1.46)	16,130 (1.58)	
Congestive heart failure *n* (%)		5345 (19.25)	161,365 (15.78)	<0.001 *
Coronary artery disease *n* (%)		6175 (22.24)	186,725 (18.27)	<0.001 *
Chronic pulmonary disease *n* (%)		8320 (29.97)	237,815 (23.26)	<0.001 *
Diabetes mellitus *n* (%)		9585 (34.53)	419,910 (41.08)	<0.001 *
Hypertension *n* (%)		19,380 (69.81)	692,565 (67.75)	<0.001 *
Renal failure *n* (%)		4790 (17.26)	150,100 (14.68)	<0.001 *
Liver disease *n* (%)		1320 (4.76)	44,360 (4.34)	0.151
Obesity *n* (%)		4665 (16.8)	267,475 (26.16)	<0.001 *
Smoking *n* (%)		10,110 (36.42)	273,260 (26.73)	<0.001 *
Alcohol abuse *n* (%)		325 (1.17)	16,275 (1.59)	0.015 *
Drug abuse *n* (%)		370 (1.33)	16,515 (1.62)	0.104
Charlson’s comorbidity index *n* (%)	0	15 (0.054)	291,175 (28.48)	<0.001 *
	1	0	0	
	2	5 (0.018)	292,840 (28.65)	
	≥3	27,740 (99.93)	438,270 (42.87)	

* Statistical significance (*p*-value < 0.05).

**Table 2 cancers-15-00222-t002:** Outcomes of patients admitted for COVID-19.

		All COVID-19 % (95%CI)	Cancer % (95%CI)	No Cancer % (95%CI)	*p*-Value
Mortality	All-cause in-hospital mortality	11.17 (10.92–11.43)	17.58 (16.56–18.66)	11.0 (10.75–11.25)	<0.001 *
Morbidity	Septic shock	3.35 (3.23–3.47)	3.87 (3.37–4.44)	3.33 (3.21–3.46)	0.032 *
	Acute respiratory failure	56.12 (55.48–56.76)	56.93 (55.5–58.36)	56.1 (55.46–56.74)	0.221
	ARDS	5.22 (5.0–5.44)	4.99 (4.42–5.63)	5.22 (5.01–5.45)	0.433
	Mechanical ventilation	9.38 (9.19–9.58)	10.14 (9.36–10.98)	9.36 (9.16–9.56)	0.508
Resource utilization	Mean LOS (days)	7.48 (7.41–7.55)	8.07 (7.83–8.31)	7.46 (7.4–7.53)	<0.001 *
	Mean total hospitalization charges (USD)	78,591 (76,384–80,798)	82,894 (77,966–87,823)	78,474 (76,269–80,679)	0.051

Abbreviations: CI; confidence interval, ARDS; acute respiratory distress syndrome, LOS; length of hospital stay. * Statistical significance (*p*-value < 0.05).

**Table 3 cancers-15-00222-t003:** Outcomes of the multivariate regression adjusted analysis.

		Adjusted Outcomes (95%CI)	*p*-Value
Mortality aOR (95%CI)	All-cause in-hospital mortality	1.21 (1.12–1.31)	<0.001 *
Morbidity aOR (95%CI)	Septic shock	0.95 (0.8–1.12)	0.524
	Acute respiratory failure	1.14 (1.06–1.22)	<0.001 *
	ARDS	0.93 (0.8–1.07)	0.31
	Mechanical ventilation	0.98 (0.88–1.09)	0.767
Resource utilization aMD (95%CI)	LOS (days)	−0.1 (−0.39–0.18)	0.479
	Total hospitalization charges (USD)	136 (−5323–5594)	0.961

Abbreviations: aOR; adjusted odds ratio, ARDS; acute respiratory distress syndrome, aMD; adjusted mean difference, LOS; length of hospital stay. Variables were adjusted for were age, sex, race, admission type, admission day, hospital bed size, hospital teaching status, hospital region, insurance status, median annual income, and comorbidities. * Statistical significance (*p*-value < 0.05).

## Data Availability

This publication is based on research using de-identified individual participant data using the Healthcare Cost and Utilization Project Nationwide Inpatient Sample (NIS) 2020 database. The authors declare that all the data supporting the findings of this study are available within the manuscript.

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
