# Peer review of "In-Hospital Mortality and Morbidity in Cancer Patients with COVID-19: A Nationwide Analysis from the United States"

_cancers, 2022, doi:10.3390/cancers15010222_

Round 1
Reviewer 1 Report
This is a well-written and well-structured paper.
Some suggestions:
1) Please try to promote better the value of the comparison between the 2 significant unbalanced groups of patients (cancer yes/no). I am not convinced yet about the importance of your results because you compare 2 irrelevant groups of patients. Cancer patients are quite less and have significant worse health status. You may consider to examine the differences in mortality, morbidities and resources usage for the different types of cancer patients, rather than the non-cancer.
2) Please check the misprint on Table 1 in "Drug abuse" row "n (%) 370 (1.330"
Author Response
Dear reviewer,
Thank you very much for your suggestions and feedback. We really appreciated them to improve our manuscript. Please see below our responses.
Comment 1: Please try to promote better the value of the comparison between the 2 significant unbalanced groups of patients (cancer yes/no). I am not convinced yet about the importance of your results because you compare 2 irrelevant groups of patients. Cancer patients are quite less and have significant worse health status. You may consider to examine the differences in mortality, morbidities and resources usage for the different types of cancer patients, rather than the non-cancer.
Response 1: We sincerely thank you for the feedback and suggestions. As we mentioned in the introduction section (lines 58–59), cancer patients are considered a vulnerable population and may be at higher risk for complications due to COVID-19. This can be related to the cancer pathophysiology or related to the other comorbidities present in cancer patients. We agree that cancer patients usually have significantly worse health status and co-morbidities compared to non-cancer patients. We adjusted for potential confounding factors between cancer and non-cancer groups to address this and also to address the difference in the number of cancer and non-cancer patients in our study. These factors included sociodemographics (age, sex, race, and median annual income), common comorbidities (congestive heart failure, coronary artery disease, chronic pulmonary disease, diabetes mellitus, hypertension, renal failure, obesity, smoking, and alcohol and drug abuse), and hospital characteristics (admission type, admission day, hospital bed size, hospital teaching status, hospital region, and insurance status). We described the adjustment in the methods section, under study variables and outcomes (lines 103 – 109) and statistical analysis (lines 124–125). Also, we mentioned the confounders as a note under Table 3 and figure 2, and we referred to the odds ratio and mean difference as “adjusted odds ratio” and “adjusted mean difference” respectively throughout our manuscript.
We'd like to thank you once more for suggesting that we look into the differences in outcomes for different types of cancer patients. We performed a sensitivity analysis of mortality outcome by cancer type (lung, breast, colorectal, prostate, leukemia, lymphoma, and myeloma) and found that lung cancer patients have the worst mortality outcome. We highlighted the results from our sensitivity analysis in a forest plot (Figure 2). In addition, we described the morbidity and resource utilization outcomes in different types of cancer in supplementary table 2, as we chose to focus on mortality as the primary outcome in the main text. We revised and edited our manuscript and added a comment about the outcomes according to cancer types. This is reflected in lines 202–204 in our manuscript.
Comment 2: Please check the misprint on Table 1 in "Drug abuse" row "n (%) 370 (1.330"
Response 2: Thank you very much for this note. This was edited and changed in Table 1.
Thank you once again, we really appreciate the feedback to improve our manuscript.
Author Response
Dear Reviewer,
We would like to thank the reviewer for the feedback and comments on our manuscript. We revised and edited our manuscript according to the suggestions as the following:
M1: Thank you for your note. The space was removed and replaced with (³). Please see line 32.
M2: Thank you for this. The DOI was added. Please see ref 7.
M3 and M4: Thank you for your comments. The access dates were added. Please see ref 10 and 11.
M5: Thank you for your note. The DOI was added. Please see ref 12.
M6: DOI was added, please see ref 17. Thank you for the comment.
M7: Thank you very much for your feedback. We added the appropriate article to our manuscript. Please refer to ref 20. Also, this is a link to the article
(https://link.springer.com/article/10.1007/s00277-020-04328-4)
M8: DOI was added. Please see ref 21. Thank you for your comment.
M9: Thank you for this. DOI was added, please see ref 24.
Thank you once again for helping in improving our manuscript.
Reviewer 3 Report
This is a well written and interesting study that demonstrates higher all cause mortality among COVID 19 patients. Remarkably, patients with lung cancer had the worst outcome. I suggest to the authors to make on the discussion a comment on the role of primary care to the COVID 19 pandemic, in the prevention of hospitalization among patients with cancer (primary care and the covid 19 pandemic by Symvoulakis EK)
Author Response
Dear Reviewer,
Thank you very much for your suggestions and feedback. We really appreciate it. Please see below our response to your comment.
Comment: This is a well written and interesting study that demonstrates higher all cause mortality among COVID 19 patients. Remarkably, patients with lung cancer had the worst outcome. I suggest to the authors to make on the discussion a comment on the role of primary care to the COVID 19 pandemic, in the prevention of hospitalization among patients with cancer (primary care and the covid 19 pandemic by Symvoulakis EK)
Response:
We sincerely thank you, and we appreciate the feedback, comments, and help in improving our manuscript. We agree that primary care service has an essential role during the COVID-19 pandemic. We added a comment to reflect this in our manuscript. Please see lines 232 – 234 in the discussion part. Thanks once again for the suggestion.
Round 2
Reviewer 1 Report
No more comments from my side.